# Pre-outbreak determinants of perceived risks of corona infection and preventive measures taken. A prospective population-based study

**Peter G. van der Velden**[1,2]*, **Miquelle Marchand**[1], **Boukje Cuelenaere**[1], **Marcel Das**[1,2,3]

**1** CenterData, Tilburg, The Netherlands, **2** Tilburg University's Network on Health and Behavior (Nethlab), Tilburg, The Netherlands, **3** Tilburg School of Economics and Management, Tilburg University, Tilburg, The Netherlands

* pg.vandervelden@tilburguniversity.edu, pg.vandervelden@kpnmail.nl

**Data Availability Statement:** The study was conducted using the Dutch Longitudinal Internet studies for the Social Sciences (LISS) panel [13].

## Abstract

### Objectives

Assess how people perceive the risks of coronavirus infection, whether people take preventive measures, and which pre-outbreak factors contribute to the perceived risks and measures taken, such as pre-outbreak respiratory problems, heart problems, diabetes, anxiety and depression symptoms, loneliness, age, gender, marital and employment status and education level.

### Methods

Data were collected in the longitudinal LISS panel, based on a random sample of the Dutch population. The coronavirus survey started on March 2, and the data collection ended on March 17 2020. Data were linked with surveys on health and social integration conducted at the end of 2019 (N$^{\text{study sample}}$ = 3,540).

### Results

About 15% perceived the risk of infection as high, and 11% the risk becoming ill when infected. Multivariable logistic regression analyses showed the following. Older age-groups perceived the risk for coronavirus infection as lower (all adjusted Odd Ratio's [aOR] ≤ .070). In total, 43.8% had taken preventive measures, especially females (aOR = 1.46, 95% CI = 1.26–1.70). Those with lower education levels less often used preventive measures (aOR = 0.55, 95% CI = 0.45–0.67). Those with pre-outbreak respiratory problems (aOR = 2.75, 95% CI = 2.11–3.57), heart problems (aOR = 1.97, 95% CI = 1.34–2.92) and diabetes (aOR = 3.12, 95% CI = 2.02–4.82) perceived the risk becoming ill when infected as higher than others. However, respondents with pre-outbreak respiratory problems and diabetes did not more often take preventive measures.

### Conclusions

Vulnerable patients more often recognize that they are at risk becoming ill when infected by the coronavirus, but many do not take preventive measures. Interventions to stimulate the

The LISS panel started in 2007 and is based on a large traditional probability sample drawn from the Dutch population. The Netherlands Organization for Scientific Research funded the set-up of LISS. Panel members receive an incentive of €15 per hour for their participation and those who do not have a computer and/or Internet access are provided with the necessary equipment at home. Further information about all conducted surveys and regulations for free access to the data can be found at www.lissdata.nl (in English). The LISS panel has received the international Data Seal of Approval (see https://www.datasealofapproval.org/en/). All data of studies conducted with the LISS panel are anonymized. Data on corona-related questions will be added to the open access data archive soon.

**Funding:** The authors received no specific funding for this work.

**Competing interests:** The authors have declared that no competing interests exist.

use of preventive measures should pay additional attention to physically vulnerable patients, males and those with lower education levels.

## Introduction

On December 31 2019, the WHO China Country Office was informed of cases of pneumonia with a then unknown etiology. The Chinese authorities identified the etiology: a new type of corona virus (SARS-CoV-2) which was isolated on January 7 [1]. In the first two months after the first report, 79,968 persons in China were infected by the virus (confirmed cases) [2]. The number of confirmed cases across the globe on March 1 2020 was raised to 87,137. With respect to the spectrum of the disease COVID-19 caused by the new corona virus, Wu and McGoogan [3] reported that, based on the 44,415 confirmed cases in China, 81% was mild, 14% severe and 5% critical. The overall case-fatality rate (CFR) in China was 2.3% (among 44,472 confirmed cases). Meanwhile, the corona virus outbreak also severely affects the production facilities, transport, the global economy, and financial markets.

To prevent and reduce infection by the new coronavirus health organizations such as the WHO, governmental health agencies and journals offer information about possible preventive measures [1–5]. The cohort study of Pan and colleagues [6] among 32,583 confirmed COVID-19 cases in Wuhan, reported between December 2019 and March 8 2020, showed that series of multifaceted (preventive) public health interventions were temporally associated with improved control over the SARS-CoV-2 outbreak. These interventions were aimed at control of the sources of infection medical resources, patient triage), blocking of transmission routes (intracity and intercity transportations, social distancing) and prevention of new infections (personal hygiene, home confinement, health communication).

To target and implement interventions to stimulate preventive behavior against infection, more insight is needed in how people perceive the risks of being infected by this new coronavirus, if they use of preventive measures, and especially which pre-outbreak factors determine the perceived risks and measures taken [7]. The study of Wang and colleagues [8], using a snowball sampling strategy in mainland China with surveys at the end of January and the end of February 2020, showed that 11.2% (first survey) and 9.1% (second survey) did find it very likely contracting COVID-19 during the pandemic. In addition, 11.9% (first survey) and 8.9% (second survey) did find it not very likely or not likely at all surviving if infected by COVID-19. Both variables were associated with current anxiety or depression symptoms. In total, 59.8% (first survey) and 73.2% (second survey) did always wear facemasks regardless of the presence or absence of symptoms; 66.6% (first survey) and 73.9% (second survey) did always wash hands after touching contaminated objects. The frequency of used preventive measures was negatively associated with current mental health problems. However, to the best of our knowledge, to date prospective studies conducted among random samples of the general population assessing the perceived risks of corona infection, preventive measures taken and pre-outbreak determinants of perceived risks and measures taken, are absent. Aim of the present prospective study, based on a random sample of the general population, is to shed light on this gap of scientific knowledge.

With respect to perceived risks, we made a distinction between risk for infection and risk of becoming ill when infected [9]. With respect to potential determinants, we first focused on pre-outbreak respiratory, heart problems and diabetes because they increase the risk for severe health problems when infected [10]. We furthermore assessed pre-outbreak anxiety and

depression symptoms, and loneliness because they may impact the perceived threat of infection and perceived likelihood to become ill when infected [11–14]. We assessed demographics such as age and gender because older people and males are more at risk to become ill [6,7]. We finally assessed pre-outbreak employment status such as having paid employment, being a job seeker or student, and having a (partial) work disability because, although employment status is associated with mental health, the extent to which employment status is associated with perceived risks and preventive measure taken is unknown. This study is conducted in the Netherlands and during the data collection period (March 2-March 17, 2020), the number of confirmed cases in the Netherlands increased rapidly from 10 to 1715 and 43 infected people (confirmed cases) died until March 17.

## Materials and methods

### Procedures and participants

The study was conducted using the Dutch Longitudinal Internet studies for the Social Sciences (LISS) panel [15]. The LISS panel started in 2007 and is based on a large traditional probability sample drawn from the Dutch population. The Netherlands Organization for Scientific Research funded the set-up of LISS. Panel members receive an incentive of €15 per hour for their participation and those who do not have a computer and/or Internet access are provided with the necessary equipment at home.

Further information about all conducted surveys and regulations for free access to the data can be found at www.lissdata.nl (in English). The LISS panel has received the international Data Seal of Approval (see https://www.datasealofapproval.org/en/). All data of studies conducted with the LISS panel are anonymized. Data on corona-related questions will be added to the open access data archive soon.

The data collection with respect to the coronavirus started on March 2 2020 (T2). Because of the rapid developments of the corona outbreak, we choose to use the data collected until March 17 2020 11.00 AM ($N^{invited}$ = 6,735, response = 70.1%). A reminder was send on the 10th day.

Data on physical and mental health problems and loneliness of the respondents before the corona outbreak were extracted from two surveys conducted at the end of 2019. These are Social Integration and Leisure survey (T1[a]; conducted in October-November 2019, $N^{invited}$ = 5,929, response = 84.2%) and the Health survey (T1[b]; conducted in November-December 2019, $N^{invited}$ = 5,954, response = 86.4%). The data of the three surveys were linked and in total 3,540 adult respondents participated in all three surveys.

We furthermore assessed 16 exclusive demographic profiles among the total adult Dutch population 2019 ($N^{2019}$ = 13,926,066), based on data of Statistics Netherlands. The 16 profiles were constructed using the following demographic characteristics: gender (2 categories), age categories (4 categories) and marital status (2 categories) totaling 2*4*2 = 16 exclusive demographic profiles. In case a profile in our study sample differed from the general population, a weighting factor was computed and applied. All results are based on the weighted sample and across tables; total numbers may slightly differ because of the weighting.

### Ethical approval and informed consent

According to the Dutch Medical Research Involving Human Subjects Act (WMO) the present study did not require ethical approval. In accordance with the General Data Protection Regulation, participants gave explicit consent for the use of the collected data for scientific and policy relevant research.

## Measures

**Perceived risk corona infection.** The Corona survey (T2) started with the following brief introduction "The next question are about the new corona virus. There is currently an outbreak of this virus in China. Now, also people in the Netherland and in other countries have become ill".

We administered two questions, developed for this study, to gain insight in how adults perceived the risks of the coronavirus. Respondents were asked: What do you think is the chance that you . . . in the next two months?: 1.) become infected with this coronavirus, and 2.) get severely ill, if you become infected with this coronavirus. Both questions had a 7-points answer scales (see Table 2).

**Preventive measures against corona.** After completing these questions, respondents were asked "In the past two months did you do things to prevent infection by this coronavirus as much as possible? (1 = yes, 2 = no)". In case respondents answered "yes", they were asked to indicate what they exactly did. The answer categories were (partly) based on WHO recommendations ((1 = the purchase of mouth masks, 2 = wash hand more often and longer, 3 = not going to certain (busy) places, 4 = cancelled a journey, 5 = otherwise, namely, (open answer category)). When respondents answered "no", they were asked why not (1 = because I do not know what I should do, 2 = but maybe I will do this still, 3 = because I have not thought about it yet, 4 = because I find it nonsense, 5 = because, namely; open answer category). For both questions respondents could choose for more than one answer.

**Pre-outbreak physical health problems.** The Health survey (T1[b]) assessed several Physician-diagnosed Diseases (PD) in the past year (1 = yes, 2 = no) and Health Problems (HP) respondents regularly suffer from (1 = yes, 0 = no). For the present study we focused on reported: 1.) respiratory problems ((PD = chronic lung disease such as chronic bronchitis or emphysema or asthma) or (HP = short of breath, problems with breathing, or coughing, a stuffy nose or flu-related complaints)); 2.) heart problems ((PD = angina, pain in the chest a heart attack including infarction or coronary thrombosis or another heart problem including heart failure) or (HP = heart complaints or angina, pain in the chest due to exertion); and 3.) diabetes (PD = diabetes or a too high blood sugar level).

**Pre-outbreak loneliness.** Loneliness at T1[a] was assessed using the six-item De Jong Gierveld Loneliness Scale (Cronbach's Alpha = .85) [16]. Respondents are asked to rate items such as 'I often feel deserted' and 'there are enough people I can count on in case of a misfortune' on three-point Likert scales (1 = yes, 2 = more or less, 3 = no). We calculated the total score after recoding the three negative formulated items and lower scores reflect more loneliness. For the present study we dichotomized scores into low ($\geq$ 15) and high loneliness ($\leq$ 14). About 20% of the respondents have scores of 14 or lower (two lowest percentiles).

**Pre-outbreak anxiety and depression symptoms.** Anxiety and depressive symptoms in the past months were examined at T1[b] using the 5-item Mental Health Index or Inventory (MHI-5) [17, 18]. The MHI-5 ask respondents to rate the presence of symptoms during the past month on 6-point Likert scales (1 = never to 6 = continuously). A cut-off of $\leq$ 59 was used to identify respondents with moderate to high anxiety and depression-symptom levels (Cronbach's Alpha = .86) [19].

**Demographics and employment status.** Pre-outbreak demographics and employment status (see Table 1) assessed in October-December 2020 were used in the present study.

**Participation period.** We monitored when respondents completed the corona questions. We distinguished three periods: period 1 (0–4 days after the start of the study), period 2 (5–9 days after the start of the study), and period 3 (10–15 days after the start of the study).

**Table 1. Characteristics study sample (N = 3,540).**

|  | n | % (95% CI) |
|---|---|---|
| Pre-outbreak respiratory problems |  |  |
| • no | 2,813 | 79.5 (78.1–80.8) |
| • yes | 727 | 20.5 (19.2–21.9) |
| Pre-outbreak heart problems |  |  |
| • no | 3,317 | 93.7 (92.9–94.5) |
| • yes | 223 | 6.3 (5.5–7.1) |
| Pre-outbreak diabetes |  |  |
| • no | 3,385 | 95.6 (94.9–96.2) |
| • yes | 155 | 4.4 (3.8–5.1) |
| Pre-outbreak anxiety and depression symptoms |  |  |
| • no | 2,785 | 78.7 (77.3–80.0) |
| • yes | 755 | 21.3 (20.0–22.7) |
| Pre-outbreak loneliness |  |  |
| • no | 2,754 | 77.8 (76.4–79.1) |
| • yes | 786 | 22.2 (20.9–23.6) |
| Age (in years) |  |  |
| • 65 or older | 944 | 26.7 (25.2–28.1) |
| • 50–64 | 837 | 23.6 (22.3–25.1) |
| • 35–49 | 916 | 25.9 (24.5–27.3) |
| • 18–34 | 843 | 23.8 (22.4–25.2) |
| Gender |  |  |
| • male | 1,744 | 49.3 (47.6–50.9) |
| • female | 1,796 | 50.7 (49.1–52.4) |
| Education |  |  |
| • high | 1,459 | 41.2 (39.6–42.8) |
| • medium | 1,277 | 36.1 (34.5–37.7) |
| • low | 803 | 22.7 (21.3–24.1) |
| Married |  |  |
| • no | 1,705 | 48.2 (46.5–49.8) |
| • yes | 1,835 | 51.8 (50.2–53.5) |
| Employment status |  |  |
| • paid employment | 1,786 | 50.5 (48.8–52.1) |
| • self-employed | 198 | 5.6 (4.9–6.4) |
| • job seeker | 73 | 2.1 (1.6–2.6) |
| • student | 278 | 7.9 (7.0–8.8) |
| • takes care of housekeeping | 256 | 7.2 (6.4–8.1) |
| • pensioner | 675 | 19.1 (17.8–20.4) |
| • has (partial) work disability | 154 | 4.4 (3.7–5.1) |
| • other | 120 | 3.4 (2.8–4.0) |
| Period participation |  |  |
| • day 1–5 | 1,844 | 52.1 (50.4–53.7) |
| • day 6–10 | 508 | 14.4 (13.2–15.5) |
| • day 11–15 | 1,188 | 33.6 (32.0–35.1) |

95% CI = 95% Confidence Interval. Results based on weighted data (gender, age, marital status).

[1]Education level: high = higher professional education/university, medium = higher general secondary/pre-university education, intermediate professional education. low = primary education, preparatory intermediate vocational education, or other.

## Data analyses

Chi-square tests and multivariable logistic regression analyses were conducted with pre-outbreak medical health problems, symptoms, loneliness, demographics, employment status, and participation period as predictors, and perceived risks and measures taken as dependent variables. Due to low cell counts in the extremes of perceived risks (see Table 1), we recoded the perceived risks into the following three categories. To optimize readability, hereafter we label these three categories of perceived risks as low (no to small chance), medium (between small and big chance) and high (big chance to absolute certain). After this recoding we assessed to what extent the predictors were associated with the perceived medium and high risk.

A similar strategy was used to assess which factors were associated with whether respondents took preventive measures.

People may perceive the risks as high and therefore take measures, but the opposite may also be true. People may perceive the risk as lower because they take measures. Since the perceived risks and preventive measures taken were assessed at the same time, we therefore did not add the perceived risk to the list of predictors in the multivariable logistic regression analyses predicting preventive measures taken.

All analyses were conducted with IBM SPSS version 26.

## Results

### Characteristics respondents

Table 1 provides an overview of the characteristics of the weighted study sample, e.g. the prevalence of pre-outbreak health problems, symptoms, loneliness, demographics, and employment status. The increase in respondents after day 9 can be attributed to the reminder mail.

### Perceived risk of infection and illness

In Table 2 shows that a minority (15.0%) perceived the risk of being infected as high. A somewhat lower proportion perceived the risk for becoming ill when infected as high (10.6%). On the other hand, very few respondents perceived the risk of infection and becoming ill as zero (4.4% and 5.5% respectively).

### Predictors perceived risk of infection corona

The results of the chi-square test and the stepwise multivariable regression analyses are presented in Table 3. We focus on the results on the stepwise regression analyses (adjusted Odds Ratios). They show that respondents with pre-outbreak heart problems more often perceive the risk of infection as medium and high than respondents without these health problems. Anxiety and depression symptoms and loneliness were not independently associated with the perceived risk. Older and low educated respondents less often perceived the risk of infection as high than younger respondents and higher educated respondents respectively. Respondents who participated later, more often perceived the risk of infection as high than those who participated in the first 4 days. Females more often than males perceived the risk of infection as medium. Those with paid employment did not more often perceive the risk as medium or high than the other employment categories, except students who less perceived the risk as a medium risk. Respondents who participated later more often perceived the risk of infection as medium and high.

### Predictors perceived risk for becoming ill when infected

Table 4 contains the results of the same analyses but with the perceived risk for becoming ill when infected in the next two months as dependent variable (right side). On a bi-variate level,

**Table 2. Perceived risks and preventive measures regarding coronavirus (N = 3,540).**

| | n | % (95% CI) |
|---|---|---|
| Perceived risk infected by corona next 2 months | | |
| • no chance | 156 | 4.4 (3.8–5.1) |
| • very small chance | 768 | 21.7 (20.4–23.1) |
| • small chance | 1,064 | 30.1 (28.6–31.6) |
| • between small and big chance | 1,018 | 28.8 (27.3–30.3) |
| • big chance | 393 | 11.1 (10.1–12.2) |
| • very big chance | 115 | 3.2 (2.7–3.9) |
| • absolutely certain | 26 | 0.7 (0.5–1.1) |
| Perceived risk will become ill when infected by corona in next 2 months | | |
| • no chance | 195 | 5.5 (4.8–6.3) |
| • very small chance | 996 | 28.1 (26.7–29.6) |
| • small chance | 1,222 | 34.5 (33.0–36.1) |
| • between small and big chance | 756 | 21.3 (20.0–22.7) |
| • big chance | 271 | 7.7 (6.8–8.6) |
| • very big chance | 73 | 2.1 (1.6–2.6) |
| • absolutely certain | 28 | 0.8 (0.5–1.1) |
| Taken measures to prevent corona infection | | |
| • no | 1,988 | 56.2 (54.5–57.8) |
| • yes | 1,552 | 43.8 (42.2–45.5) |

95% CI = 95% Confidence Interval. Results based on weighted data (gender, age, marital status).

almost all predictors were significantly associated. The multivariable analyses showed that respondents with pre-outbreak physical health problems, anxiety and mental health problems and loneliness, more often perceived the risk for becoming ill when infected as high than others. Older respondents more often, in contrast to the perceived risk of infection, perceived the risk for becoming ill as medium and high than younger respondents.

## Preventive measures taken and predictors

Of the total study sample, 43.8% took preventive measures (see Table 2) such as washing hands more often and longer (92.2%), not going to work of avoid certain (busy) places (53.6%), purchase of mouth masks (5.9%) and cancelled a journey (8.2%). Of the respondents who did not take preventive measures, 42.5% reported that they find it nonsense or useless, 24.9% that maybe will do this still, 20.4% have not thought about it yet, and 15.4% that they do not know what they should do.

Table 5 shows which factors predicted the use of preventive measures against infection by the coronavirus. With respect to pre-outbreak physical health problems: only respondents with heart problems took preventive measures more often. Females more often took preventive measures, and medium and high educated respondents more often than low educated respondents. Finally, respondents who filled in the survey more recently, more often took preventive measures. With respect to employment status, no differences were found between respondents with paid employment and all other employment categories.

We repeated the regression analyses among those who participated 10–15 days after the start of the corona survey, showing almost similar results. Having heart problems was no longer significantly associated with preventive measures, while respondents in the age category 35–49 years old more often took preventive measures than the youngest subgroup of respondents.

**Table 3. Predictors of perceived risk of corona infection (N = 3,540).**

| | | Low risk become infected in next two months versus | | | | |
|---|---|---|---|---|---|---|
| | | Medium risk will become infected | | | High risk will become infected | |
| | n | % medium | aOR (95% CI) | n | % high | aOR (95% CI) |
| Pre-outbreak respiratory problems | | | | | | |
| • no (ref.) | 2,396 | 32.9* | 1 | 2,024 | 20.6 | 1 |
| • yes | 609 | 37.4 | 1.26 (1.03–1.54)* | 499 | 23.6 | 1.32 (1.00–1.76) |
| Pre-outbreak heart problems | | | | | | |
| • no (ref.) | 2,818 | 33.5 | 1 | 2,373 | 21.0 | 1 |
| • yes | 188 | 39.4 | 1.42 (1.02–1.98)* | 149 | 23.5 | 2.70 (1.67–4.35)*** |
| Pre-outbreak diabetes | | | | | | |
| • no (ref.) | 2,861 | 34.0 | 1 | 2,413 | 21.7** | 1 |
| • yes | 144 | 31.3 | 1.05 (0.72–1.55) | 110 | 10.0 | 0.63 (0.31–1.28) |
| Pre-outbreak anxiety and depression symptoms | | | | | | |
| • no (ref.) | 2,385 | 32.5** | 1 | 2,010 | 20.0 | 1 |
| • yes | 621 | 39.0 | 1.20 (0.97–1.47) | 513 | 26.1** | 1.11 (0.84–1.47) |
| Pre-outbreak loneliness | | | | | | |
| • no (ref.) | 2,353 | 33.5 | 1 | 1,965 | 20.4 | 1 |
| • yes | 653 | 35.1 | 0.95 (0.78–1.17) | 558 | 24.0 | 1.16 (0.88–1.52) |
| Age (in years) | | | | | | |
| • 18–34 (ref.) | 719 | 37.1*** | 1 | 677 | 33.2*** | 1 |
| • 35–49 | 664 | 36.4 | 0.70 (0.54–0.90)** | 595 | 29.1 | 0.61 (0.45–0.83)** |
| • 50–64 | 819 | 35.0 | 0.67 (0.52–0.87)** | 630 | 15.6 | 0.29 (0.21–0.41)*** |
| • 65 or older | 804 | 27.5 | 0.48 (0.32–0.73)** | 622 | 6.3 | 0.11 (0.05–0.22)*** |
| Gender | | | | | | |
| • male (ref.) | 1,485 | 29.2*** | 1 | 1,312 | 19.8 | 1 |
| • female | 1,521 | 38.5 | 1.57 (1.33–1.85)*** | 1,211 | 22.7 | 1.18 (0.94–1.48) |
| Education level | | | | | | |
| • high (ref.) | 1,180 | 33.6 | 1 | 1,061 | 26.2*** | 1 |
| • medium | 1,097 | 35.7 | 1.10 (0.92–1.33) | 885 | 20.3 | 0.65 (0.51–0.84)** |
| • low | 728 | 31.3 | 1.02 (0.81–1.27) | 576 | 13.2 | 0.60 (0.43–0.84)** |
| Married | | | | | | |
| • yes (ref.) | 1,480 | 34.8 | 1 | 1,190 | 18.9** | 1 |
| • no | 1,526 | 33.0 | 0.81 (0.68–0.97)* | 1,332 | 23.2 | 0.78 (0.61–1.00) |
| Employment status | | | | | | |
| • paid employment | 1,465 | 37.2*** | 1 | 1,241 | 25.9*** | 1 |
| • self-employed | 171 | 30.4 | 0.73 (0.51–1.05) | 126 | 18.5 | 0.75 (0.46–1.22) |
| • job seeker | 66 | 43.9 | 1.37 (0.81–2.32) | 44 | 15.9 | 0.73 (0.29–1.79) |
| • student | 205 | 30.2 | 0.63 (0.44–0.91)* | 216 | 33.8 | 1.24 (0.83–1.83) |
| • housekeeping | 222 | 35.1 | 0.78 (0.55–1.10) | 179 | 19.6 | 1.08 (0.66–1.76) |
| • pensioner | 643 | 27.8 | 0.97 (0.65–1.44) | 496 | 6.5 | 1.04 (0.49–2.20) |
| • (partial) work disab. | 128 | 37.5 | 0.93 (0.62–1.41) | 106 | 24.5 | 1.41 (0.82–2.43) |
| • other | 107 | 24.3 | 0.56 (0.35–0.92)* | 94 | 13.8 | 0.60 (0.30–1.20) |
| Period participation | | | | | | |
| • 0–4 days (ref.) | 1,699 | 25.9*** | 1 | 1,404 | 10.3*** | 1 |
| • 5–9 days | 451 | 31.9 | 1.37 (1.09–1.72)** | 364 | 15.7 | 2.00 (1.41–2.85)*** |

*(Continued)*

**Table 3.** (Continued)

| | | | Low risk become infected in next two months versus | | | |
| --- | --- | --- | --- | --- | --- | --- |
| | | | Medium risk will become infected | | High risk will become infected | |
| | n | % medium | aOR (95% CI) | n | % high | aOR (95% CI) |
| • 10–15 days | 855 | 50.8 | 3.03 (2.54–3.62)*** | 754 | 44.2 | 7.76 (6.09–9.90)*** |

aOR = Odds Ratio adjusted for all other variables in table. 95 CI = 95% confidence interval of aOR. Ref = reference category. Low risk = no to small chance (n = 1,988). Medium risk = between small and big chance (n = 1,018). High risk = big chance to absolute certain (n = 535). Results based on weighted data (gender, age, marital status). housekeeping = takes care of housekeeping. (partial) work disab. = has (partial) work disability. The asterisks near the percentages refer to the p-values of the chi-square tests, and the asterisks near the 95% CI's refer to the p-values of the aOR's.

* p < .05

** p < .01

*** p < .001.

## Discussion

Main results of this prospective population based-study are that during the 2-week study period (March 2 to March 17 2020) the number of respondents who perceived the risk of being infected by the new coronavirus SARS-CoV-2 as high, increased sharply (10% to 44%). Multivariable logistic regression analyses showed that respondents with pre-outbreak respiratory and heart problems, diabetes, anxiety and depression symptoms and loneliness, and older respondents more often perceived the risk becoming ill when infected as high. Although older respondents compared to the youngest respondents less often perceived the risk of being infected as high, compared to the youngest adults they more often perceived the risk of becoming ill when infected as high. The last finding is in line with the general information provided by governmental health agencies and media before and during our study period, suggesting that this information reached these specific groups. In line with the increased perceived risk to be infected, the number of respondents who took preventive measures increase too. However, respondents with pre-outbreak respiratory problems and diabetes did not more often take preventive measures than others, although they perceived the risk of becoming ill when infected more often as high. A similar remarkable pattern was found for pre-outbreak loneliness and anxiety and depression symptoms. In addition, analyses of respondents who participated 10–15 after the start of the study showed that respondents with respiratory problems, heart problems and diabetes did not differ in the proportion of people who took preventive measures. With respect to employment status, the multivariable logistic regression analyses furthermore showed that students more often perceived the risk of infection as medium, but not more often as high compared to respondents with paid employment. Respondents with (partial) work disabilities compared to those with paid employment, more often perceived the risk of infection and becoming ill when infected as medium and high. Nevertheless, those with paid employment did not differ in the prevalence of preventive measures taken from the other employment subgroups.

Our findings are somewhat similar to the results of a study reported by the WHO Regional Office for Europe [6]. This serial cross-sectional study conducted in Germany in almost the same period as our study (week 10 and 11 2020) showed that the prevalence of respondents who perceived the risk to be infected by the coronavirus as high, increased from 16.8% to 21.4%. They furthermore reported, like us, that older respondents (60+) felt less likely be infected. In the study by Wang and colleagues [8] about 10% did not found it very likely or not likely at all to survive COVID-19. We have no data to compare these findings with. Importantly, in our study the effects of other factors that are associated with the perceived risk of

**Table 4. Predictors of perceived risk to become ill when infected by coronavirus (N = 3,540).**

| | | Low risk will become ill in next two months versus | | | | |
|---|---|---|---|---|---|---|
| | | Medium risk will become ill | | | High risk will become ill | |
| | n | % medium | aOR (95% CI) | n | % high | aOR (95% CI) |
| Pre-outbreak respiratory problems | | | | | | |
| • no (ref.) | 2,603 | 22.2*** | 1 | 2,235 | 9.4*** | 1 |
| • yes | 564 | 31.4 | 1.42 (1.15–1.77)** | 549 | 29.5 | 2.75 (2.11–3.57)*** |
| Pre-outbreak heart problems | | | | | | |
| • no (ref.) | 3,014 | 23.5* | 1 | 2,609 | 11.6*** | 1 |
| • yes | 154 | 31.2 | 0.96 (0.66–1.41) | 175 | 39.4 | 1.97 (1.34–2.92)** |
| Pre-outbreak diabetes | | | | | | |
| • no (ref.) | 3,062 | 23.4* | 1 | 2,667 | 12.1*** | 1 |
| • yes | 105 | 35.2 | 1.30 (0.85–1.99) | 117 | 41.9 | 3.12 (2.02–4.82)*** |
| Pre-outbreak anxiety and depression symptoms | | | | | | |
| • no (ref.) | 2,537 | 22.7** | 1 | 2,211 | 11.3*** | 1 |
| • yes | 631 | 28.7 | 1.31 (1.04–1.63)* | 573 | 21.5 | 1.51 (1.12–2.03)** |
| Pre-outbreak loneliness | | | | | | |
| • no (ref.) | 2,508 | 22.8* | | 2,180 | 11.2*** | 1 |
| • yes | 659 | 27.6 | 1.18 (0.95–1.46) | 604 | 21.0 | 1.60 (1.21–2.13)** |
| Age (in years) | | | | | | |
| • 18–34 (ref.) | 897 | 15.5*** | 1 | 805 | 5.8*** | 1 |
| • 35–49 | 761 | 23.5 | 1.19 (0.90–1.56) | 658 | 11.6 | 1.52 (0.98–2.37) |
| • 50–64 | 803 | 25.2 | 1.22 (0.92–1.62) | 715 | 15.9 | 2.01 (1.29–3.12)** |
| • 65 or older | 708 | 33.2 | 1.17 (0.76–1.80) | 608 | 22.2 | 2.45 (1.32–4.57)** |
| Gender | | | | | | |
| • male (ref.) | 1,548 | 21.5** | | 1,411 | 13.9 | 1 |
| • female | 1,620 | 26.1 | 1.17 (0.98–1.40) | 1,373 | 12.8 | 0.86 (0.67–1.11) |
| Education | | | | | | |
| • high (ref.) | 1,326 | 19.8*** | | 1,197 | 11.1*** | 1 |
| • medium | 1,147 | 23.5 | 1.28 (1.05–1.56)* | 1,008 | 13.0 | 1.06 (0.80–1.41) |
| • low | 696 | 32.2 | | 580 | 18.6 | 1.02 (0.74–1.41) |
| Married | | | | | | |
| • yes (ref.) | 1,513 | 27.2*** | 1 | 1,294 | 14.8* | 1 |
| • no | 1,655 | 20.8 | 0.85 (0.71–1.03) | 1,490 | 12.1 | 0.97 (0.75–1.26) |
| Employment status | | | | | | |
| • paid employment | 1,657 | 20.9*** | 1 | 1,445 | 8.9*** | 1 |
| • self-employed | 178 | 21.9 | 1.05 (0.71–1.54) | 159 | 12.6 | 1.27 (0.74–2.16) |
| • job seeker | 68 | 23.5 | 1.03 (0.57–1.86) | 58 | 10.3 | 0.81 (0.31–2.08) |
| • student | 264 | 9.8 | 0.45 (0.28–0.72)** | 252 | 5.6 | 0.81 (0.42–1.57) |
| • housekeeping | 223 | 32.3 | 1.33 (0.94–1.88) | 184 | 17.9 | 1.57 (0.94–2.62) |
| • pensioner | 565 | 34.9 | 1.90 (1.26–2.87)** | 478 | 23.0 | 1.58 (0.90–2.77) |
| • (partial) work disab. | 105 | 38.1 | 1.75 (1.13–2.70)* | 114 | 43.0 | 3.57 (2.22–5.74)*** |
| • other | 109 | 23.9 | 0.98 (0.61–1.60) | 94 | 11.7 | 0.75 (0.36–1.56) |
| Period participation | | | | | | |
| • 0–4 days (ref.) | 1,674 | 20.4*** | 1 | 1,504 | 11.4*** | 1 |
| • 5–9 days | 458 | 24.2 | 1.18 (0.91–1.51) | 396 | 12.4 | 1.24 (0.86–1.78) |

(*Continued*)

**Table 4.** (Continued)

| | | Low risk will become ill in next two months versus | | | | |
| | | Medium risk will become ill | | | High risk will become ill | |
| | n | % medium | aOR (95% CI) | n | % high | aOR (95% CI) |
| --- | --- | --- | --- | --- | --- | --- |
| • 10–15 days | 1036 | 29.2 | 1.76 (1.46–2.12)*** | 885 | 17.2 | 2.10 (1.61–2.73)*** |

aOR = Odds Ratio adjusted for all other variables in table. 95 CI = 95% confidence interval of adjusted Odds ratio. Ref = reference category. Low risk = no to small chance (n = 2,412). Medium risk = between small and big chance (n = 757). High risk = big chance to absolute certain (n = 372). Results based on weighted data (gender, age, marital status). housekeeping = takes care of housekeeping. (partial) work disab. = has (partial) work disability. The asterisks near the percentages refer to the p-values of the chi-square tests, and the asterisks near the 95% CI's refer to the p-values of the aOR's.

* p < .05

** p < .01

*** p < .001.

corona infection were controlled for such as pre-outbreak respiratory and heart problems, and education level. Asmundson and Taylor [20] reported that, according to polls, in the US 56% was very concerned about the spread of the virus and in that Canada 7% was very concerned about becoming infected. The prevalence of respondent participating in the third and last period who used preventive measures slightly approximated the prevalence found by Wang and colleagues [8].

To date many studies on our research topic are initiated and conducted. However, when finalizing this study we were unaware of studies based on random samples among the general population published in peer-reviewed journals, on the perceived risks, the use of preventive measures and their pre-outbreak determinants, to compare our findings with.

## Strengths and limitations

Strength of the present study are the use of a large traditional probability based sample drawn from the Dutch population, the prospective study-design, data on pre-outbreak physician-diagnosed diseases, and use of well validated instruments on anxiety and depression symptoms, and loneliness.

We deliberately choose to use the data that was collected in the first two weeks of the survey (response was 70.1%), to be able to share our results rapidly given the threatening global developments. However, although we distinguished three subsequent periods during these two weeks suggesting an increase in preventive measures taken, we do not know from this study if and when all respondents have taken preventive measures. In addition, we do not know from this study to what extent respondents who have taken preventive measures, will continue to comply with protection guidelines from governmental health agencies. Another limitation is that we not were able to include children. It is unknown to what extent children's perceptions of the risks and the measures they taken resembles those of adults and especially parents and other family members. We did not systematically examine whether respondents were in quarantine, e.g. were separated and restricted in movement because they had been potentially infected by the coronavirus and their effects on perceived risks [21]. The present study does not provide information on this topic, nor how quarantine affects post-quarantine preventive behavior. Finally, it was beyond the scope of the present study to assess perceived risks and preventive measures taken, as well as its pre-outbreak predictors, among (specific groups of) the workforce when returning to work after a lockdown. For this purpose, we refer to the study of Tan and colleagues [22].

**Table 5. Predictors of taken preventive measures taken in past two months (N = 3,540).**

| | | Preventive measures taken | |
|---|---|---|---|
| | n | %measures | aOR (95% CI) |
| **Pre-outbreak respiratory problems** | | | |
| • no (ref.) | 2,813 | 43.5 | 1 |
| • yes | 727 | 45.1 | 1.02 (0.85–1.23) |
| **Pre-outbreak heart problems** | | | |
| • no (ref.) | 3,317 | 43.3* | 1 |
| • yes | 224 | 51.8 | 1.53 (1.13–2.07)** |
| **Pre-outbreak diabetes** | | | |
| • no (ref.) | 3,386 | 43.9 | 1 |
| • yes | 155 | 41.3 | 0.99 (0.70–1.42) |
| **Pre-outbreak anxiety and depression symptoms** | | | |
| • no (ref.) | 2,785 | 43.4 | 1 |
| • yes | 755 | 45.6 | 1.10 (0.91–1.33) |
| **Pre-outbreak loneliness** | | | |
| • no (ref.) | 2,753 | 44.0 | 1 |
| • yes | 786 | 43.4 | 1.03 (0.86–1.24) |
| **Age (in years)** | | | |
| • 18–34 (ref.) | 944 | 39.7** | 1 |
| • 35–49 | 837 | 46.8 | 1.18 (0.94–1.47) |
| • 50–64 | 916 | 46.7 | 1.34 (1.06–1.70)* |
| • 65 or older | 843 | 42.3 | 1.39 (0.96–2.01) |
| **Gender** | | | |
| • male (ref.) | 1,744 | 39.5*** | 1 |
| • female | 1,796 | 48.1 | 1.46 (1.26–1.70)*** |
| **Education** | | | |
| • high (ref.) | 1,459 | 49.0*** | 1 |
| • medium | 1,277 | 41.8 | 0.71 (0.60–0.84)*** |
| • low | 803 | 37.6 | 0.55 (0.45–0.67)*** |
| **Married** | | | |
| • yes (ref.) | 1,705 | 45.7* | 1 |
| • no | 1,835 | 42.1 | 0.91 (0.77–1.06) |
| **Employment status** | | | |
| • paid employment | 1,786 | 44.9 | 1 |
| • self-employed | 198 | 45.5 | 0.92 (0.67–1.26) |
| • job seeker | 73 | 37.0 | 0.71 (0.42–1.20) |
| • student | 278 | 39.2 | 1.10 (0.81–1.50) |
| • housekeeping | 256 | 46.5 | 0.97 (0.71–1.32) |
| • pensioner | 675 | 41.3 | 0.89 (0.62–1.27) |
| • (partial) work disab. | 154 | 48.7 | 1.21 (0.84–1.75) |
| • other | 120 | 42.0 | 0.89 (0.59–1.36) |
| **Period participation** | | | |
| • 0–4 days (ref.) | 1,844 | 30.0*** | 1 |
| • 5–9 days | 508 | 44.5 | 1.92 (1.57–2.36)*** |

(*Continued*)

**Table 5.** (Continued)

| | | Preventive measures taken | |
|---|---|---|---|
| | **n** | **%measures** | **aOR (95% CI)** |
| • 10–15 days | 1,188 | 65.0 | 4.34 (3.70–5.08)*** |

aOR = Odds Ratio adjusted for all other variables in table. 95 CI = 95% confidence interval of aOR. Ref = reference category. Results based on weighted data (gender, age, marital status). housekeeping = takes care of housekeeping. (partial) work disab. = has (partial) work disability. The asterisks near the percentages refer to the p-values of the chi-square tests, and the asterisks near the 95% CI's refer to the p-values of the aOR's.

* p < .05

** p < .01

*** p < .001.

Nevertheless, we believe that our results are also of relevance for future SARS-CoV-2 outbreaks as well as other outbreaks.

## Future research

Future research on the perceived risks and preventive measures should, among many other important questions, focus on to what extent people continue to take the proposed or required preventive measures. Which physical, psychological, financial, and societal factors do influence compliance to (possible new) preventive measures on the medium and long term? Which interventions to stimulate constant preventive behavior are most effective? These questions are highly relevant because to date there are no indications that this pandemic will end soon. Furthermore, taken preventive measures should be assessed more in detail, and self-reports on measures taken should be complemented with peer-reports. In addition, future studies should pay special attention towards children and how they perceive the risks for coronavirus infection and if and how they protect themselves.

## Conclusions

The results of this study, based on a random sample of the general adult population, are partly reassuring and positive, and partly negative. Positive is the finding that the number of respondents who have taken preventive measures during the brief 2-weeks study period increased, while taking other significant predictors of the use of preventive measures into account. It is very likely that the daily stream of information about the pandemic and advice on this matter provided by Dutch governmental health agencies, physicians and media, contributed to this finding. A negative finding is that respondents with respiratory problems and diabetes, who are considered groups at severe risk for complicated health problems when infected, did not take preventive measures more often than others. In addition, we found no indications that people took preventive measures irrespective of their education level and gender. The last findings suggest that specific education level and gender-related interventions should be developed and offered to increase preventive behavior among men and those with a lower education level.

## Author Contributions

**Conceptualization:** Peter G. van der Velden, Miquelle Marchand, Boukje Cuelenaere, Marcel Das.

**Data curation:** Miquelle Marchand.

**Formal analysis:** Peter G. van der Velden, Marcel Das.

**Investigation:** Peter G. van der Velden, Miquelle Marchand, Boukje Cuelenaere, Marcel Das.

**Methodology:** Peter G. van der Velden, Miquelle Marchand, Boukje Cuelenaere, Marcel Das.

**Supervision:** Peter G. van der Velden.

**Validation:** Peter G. van der Velden, Miquelle Marchand, Boukje Cuelenaere, Marcel Das.

**Writing – original draft:** Peter G. van der Velden.

**Writing – review & editing:** Peter G. van der Velden, Miquelle Marchand, Boukje Cuelenaere, Marcel Das.

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
