## [Decision Letter · Decision Letter 0]

19 May 2020

PONE-D-20-09023

Pre-outbreak determinants of perceived risks of corona infection and preventive measures taken. A prospective population-based study

PLOS ONE

Dear Dr. van der Velden,

Thank you for submitting your manuscript to PLOS ONE. After careful consideration, we feel that it has merit but does not fully meet PLOS ONE’s publication criteria as it currently stands. Therefore, we invite you to submit a revised version of the manuscript that addresses the points raised during the review process.

We would appreciate receiving your revised manuscript by Jul 03 2020 11:59PM. To enhance the reproducibility of your results, we recommend that if applicable you deposit your laboratory protocols in protocols.io, where a protocol can be assigned its own identifier (DOI) such that it can be cited independently in the future. For instructions see: http://journals.plos.org/plosone/s/submission-guidelines#loc-laboratory-protocols

We look forward to receiving your revised manuscript.

Kind regards,

Geilson Lima Santana, M.D., Ph.D.

Academic Editor

PLOS ONE

Journal Requirements:

2. Please provide additional details regarding participant consent. In the ethics statement in the Methods and online submission information, please ensure that you have specified whether consent was written or verbal/oral. If consent was verbal/oral, please specify: 1) whether the ethics committee approved the verbal/oral consent procedure, 2) why written consent could not be obtained, and 3) how verbal/oral consent was recorded.

3. Please include additional information regarding the survey or questionnaire used in the study and ensure that you have provided sufficient details that others could replicate the analyses. If you developed and/or translated a questionnaire as part of this study and it is not under a copyright license more restrictive than Creative Commons Attribution (CC-BY), please include a copy, in both the original language and English, as Supporting Information.

Reviewers' comments:

Reviewer's Responses to Questions

**Comments to the Author**

1. Is the manuscript technically sound, and do the data support the conclusions?

Reviewer #1: Yes

Reviewer #2: Yes

2. Has the statistical analysis been performed appropriately and rigorously? 

Reviewer #1: No

Reviewer #2: Yes

3. Have the authors made all data underlying the findings in their manuscript fully available?

Reviewer #1: Yes

Reviewer #2: Yes

4. Is the manuscript presented in an intelligible fashion and written in standard English?

Reviewer #1: Yes

Reviewer #2: Yes

5. Review Comments to the Author

Reviewer #1: This study was based on the Dutch Longitudinal Internet studies for the Social

Sciences (LISS) panel, and collected the data with respect to the coronavirus from March 2 to March 17, 2020, aimed to assess how people perceive the risks of coronavirus infection, whether people take preventive measures, and what pre-outbreak factors contribute to the perceived risks and measures taken. They observed that the elders, males, and low educated respondents less often perceived the risk of infection. The elders and those with pre-outbreak physical health problems, anxiety and mental health problems and loneliness perceived the risk becoming ill when infected as higher than others. The subjects with pre-outbreak heart diseases, females, elders, and medium and high educated respondents more often took preventive measures.

This study was, therefore, by using a specific study population, a great opportunity to describe the current recognition of COVID-19 in Dutch populations, also could represent the other Europeans. However, a part of this manuscript needs some revisions and restructuration.

1. In the Introduction part, 1st paragraph, Line 3, “a new type of corona virus (COVID-19 or SARS-CoV-2) which was isolated on January 7” Here, the COVID-19 should be deleted. SARS-CoV-2 is the name of the corona virus named by the World Health Organization, while the disease caused by SARS-CoV-2 is designated Corona Virus Disease-19 (COVID-19). This mistake is also seen in the 2nd paragraph, Line 1. In the manuscript, the author should make correct description of the new virus and disease.

2. In the Introduction part, 1st paragraph, “The overall case-fatality rate (CFR) in China was 2.3% (among 44,472 confirmed cases).” More new data have been released in China, so the CFR could be updated. Also, the CFR is quite different between Wuhan and the other cities in China. The different number should also be described separately.

3. In the Introduction part, 2nd paragraph, “governmental health agencies and journals offer information about possible preventive measures”. A recent research published in JAMA (Pan et al, Association of Public Health Interventions With the Epidemiology of the COVID-19 Outbreak in Wuhan, China. JAMA. doi:10.1001/jama.2020.6130. Published online April 10, 2020.) have reported a series of multifaceted public health interventions taken in Wuhan, China, was temporally associated with improved control of the COVID-19 outbreak. This publication should also be added as a reference.

4. The adjusted OR(95%CI) for the factors with no significant associations with perceived risk of corona infection (Table 3), perceived risk to become ill when infected (Table 4), and taken preventive measures in past two months (Table 5), should also be added to show more information.

Reviewer #2: I have the following comments for the paper. I am happy to review the paper again.

1) Under introduction, the authors stated "peer-reviewed population-based studies". What does peer-reviewed mean?

2) This statement, " peer-reviewed population-based studies assessing the perceived risks of corona infection, measures and their determinants are absent." is incorrect. The authors need to highlight the following landmark longitudinal study that also assessed perceived risk and determinants. Please mention the findings of this study in the Introduction.

Wang C, Pan R, Wan X, et al. (2020) A Longitudinal Study on the Mental Health of General Population during the COVID-19 Epidemic in China [published online ahead of print, 2020 Apr 13]. Brain Behav Immun. 2020; S0889-1591(20)30511-0. doi:10.1016/j.bbi.2020.04.028

3) Under discussion, the authors need to have a global view and need to discuss findings beyond Germany and Canada.

For example, the authors found that "Multivariable logistic regression analyses showed that respondents with pre-outbreak anxiety and depression symptoms more often perceived the risk becoming ill when infected as high. Please refer to the following studies and discuss the challenges faced by psychiatric patients with anxiety and depression during COVID-19 lockdown.

Hao F, Tan W, Jiang L, et al. Do psychiatric patients experience more psychiatric symptoms during COVID-19 pandemic and lockdown? A Case-Control Study with Service and Research Implications for Immunopsychiatry [published online ahead of print, 2020 Apr 27]. Brain Behav Immun. 2020;S0889-1591(20)30626-7. doi:10.1016/j.bbi.2020.04.06

4) Under discussion, the authors stated "Another limitation is that we not were able to include children. It is unknown to what extent children’s perceptions of the risks and the measures they taken resembles those of adults and especially parents and other family members." Please refer to the following study from China that included participants as young as 12 year old. Students were more affected due to disruption of academic studies:

Wang C, Pan R, Wan X, et al. (2020) A Longitudinal Study on the Mental Health of General Population during the COVID-19 Epidemic in China [published online ahead of print, 2020 Apr 13]. Brain Behav Immun. 2020; S0889-1591(20)30511-0. doi:10.1016/j.bbi.2020.04.028

5) The authors should add one additional limitation. There was no mention of occupation of participants. The authors should discuss the impact of COVID on general workforce and healthcare professionals. Please discuss the findings of the following studies:

Tan W, Hao F, McIntyre RS, et al. Is Returning to Work during the COVID-19 Pandemic Stressful? A Study on Immediate Mental Health Status and Psychoneuroimmunity Prevention Measures of Chinese Workforce [published online ahead of print, 2020 Apr 23]. Brain Behav Immun. 2020;S0889-1591(20)30603-6. doi:10.1016/j.bbi.2020.04.055

Chew NWS, Lee GKH, Tan BYQ, et al. A multinational, multicentre study on the psychological outcomes and associated physical symptoms amongst healthcare workers during COVID-19 outbreak [published online ahead of print, 2020 Apr 21]. Brain Behav Immun. 2020;S0889-1591(20)30523-7. doi:10.1016/j.bbi.2020.04.049

6. PLOS authors have the option to publish the peer review history of their article (what does this mean?). If published, this will include your full peer review and any attached files.

Reviewer #1: No

Reviewer #2: Yes: Roger Ho

---

## [Author Response · Author response to Decision Letter 0]

27 May 2020

We would like to thank both reviewers very much for their time, and helpful, kind, and constructive comments. We believe that the comments enabled us to improve our paper. Below we have described in detail how we responded to each comment. 

 Both reviewers suggested new references, which we appreciated very much. The main reason that we did not refer to these studies earlier is that we submitted our original manuscript before these studies were published.

 In similar and related comments, we combined our responses. We hope that it does not inconvenience the reviewers too much. We have marked all important changes in yellow.

REVIEWER #1: 

This study was based on the Dutch Longitudinal Internet studies for the Social Sciences (LISS) panel, and collected the data with respect to the coronavirus from March 2 to March 17, 2020, aimed to assess how people perceive the risks of coronavirus infection, whether people take preventive measures, and what pre-outbreak factors contribute to the perceived risks and measures taken. They observed that the elders, males, and low educated respondents less often perceived the risk of infection. The elders and those with pre-outbreak physical health problems, anxiety and mental health problems and loneliness perceived the risk becoming ill when infected as higher than others. The subjects with pre-outbreak heart diseases, females, elders, and medium and high educated respondents more often took preventive measures.

This study was, therefore, by using a specific study population, a great opportunity to describe the current recognition of COVID-19 in Dutch populations, also could represent the other Europeans. However, a part of this manuscript needs some revisions and restructuration.

1. In the Introduction part, 1st paragraph, Line 3, “a new type of corona virus (COVID-19 or SARS-CoV-2) which was isolated on January 7” Here, the COVID-19 should be deleted. SARS-CoV-2 is the name of the corona virus named by the World Health Organization, while the disease caused by SARS-CoV-2 is designated Corona Virus Disease-19 (COVID-19). This mistake is also seen in the 2nd paragraph, Line 1. In the manuscript, the author should make correct description of the new virus and disease.

Response

Thank you for this correction. We have revised the manuscript accordingly.

2. In the Introduction part, 1st paragraph, “The overall case-fatality rate (CFR) in China was 2.3% (among 44,472 confirmed cases).” More new data have been released in China, so the CFR could be updated. Also, the CFR is quite different between Wuhan and the other cities in China. The different number should also be described separately.

Response

We understand the comment of the reviewer and we fully realize that more information is available, but we prefer to provide information in our manuscript that was available and reported in the (Dutch) media in the period before we started our study and during the period we collcted our data. 

3. In the Introduction part, 2nd paragraph, “governmental health agencies and journals offer information about possible preventive measures”. A recent research published in JAMA (Pan et al, Association of Public Health Interventions With the Epidemiology of the COVID-19 Outbreak in Wuhan, China. JAMA. doi:10.1001/jama.2020.6130. Published online April 10, 2020.) have reported a series of multifaceted public health interventions taken in Wuhan, China, was temporally associated with improved control of the COVID-19 outbreak. This publication should also be added as a reference.

Response

Thank you very much for your suggestion. The reason we did not refer to this important study is that the paper of Pan et al. was published after we submitted our paper (March 30).

Bases on this comment in the introduction section we added:

“The cohort study of Pan and colleagues [6] among 32,583 confirmed COVID-19 cases in Wuhan, reported between December 2019 and March 8 2020, showed that series of multifaceted (preventive) public health interventions were temporally associated with improved control over the SARS-CoV-2 outbreak. These interventions were aimed at control of the sources of infection medical resources, patient triage), blocking of transmission routes (intracity and intercity transportations, social distancing) and prevention of new infections (personal hygiene, home confinement, health communication).”

4. The adjusted OR(95%CI) for the factors with no significant associations with perceived risk of corona infection (Table 3), perceived risk to become ill when infected (Table 4), and taken preventive measures in past two months (Table 5), should also be added to show more information.

Response

Bases on this comment we realized that we only clarified in the results section that we conducted stepwise multivariable logistic regression analyses. That is the reason no adjusted OR’s were provided for the non-significant predictors in the Tables (because they were not entered in the regression analyses). Our apologies for this omission. 

However, based on this comment we have re-analysed our data without the stepwise procedure to be able to show the non-significant adjusted OR’s. In addition, based on comment 10 of reviewer 2, we added employment status to the list of predictors and revised the data analyses section as follows (changes in italics):

“Chi-square tests and multivariable logistic regression analyses were conducted with pre-outbreak medical health problems, symptoms, loneliness, demographics, employment status, and participation period as predictors, and perceived risks and measures taken as dependent variables”.

In addition, we revised the section on elapsed time in the measures section as follows:

Participation period 

We monitored when respondents completed the corona questions. We distinguished three periods: period 1 (0-4 days after the start of the study), period 2 (5-9 days after the start of the study), and period 3 (10-15 days after the start of the study).

In the measures section we added:

“Pre-outbreak demographics and employment status assessed in November-December 2020 were used in the present study”.

In discussion section we added:

“With respect to employment status, the multivariable logistic regression analyses furthermore showed that students more often perceived the risk of infection as medium, but not more often as high compared to respondents with paid employment. Respondents with (partial) work disabilities compared to those with paid employment, more often perceived the risk of infection and becoming ill when infected as medium and high. Nevertheless, those with paid employment did not differ in the prevalence of preventive measures taken from the other employment subgroups.”

For the revised tables, we would like to refer to the revised manuscript because the tables are rather lengthy.

In addition, we clarified the significance levels (*) a bit more in the notes under Tables 3, 4 and 5 as follows: 

“The asterisks near the percentages refer to the p-values of the chi-square tests, and the asterisks near the 95% CI’s refer to the p-values of the aOR’s”. 

REVIEWER #2: 

I have the following comments for the paper. I am happy to review the paper again.

5. Under introduction, the authors stated "peer-reviewed population-based studies". What does peer-reviewed mean?

Response

With peer-reviewed population based studies we meant studies based on random samples among the general population that were published in peer-reviewed journals. We realize that this sentence is unclear and confusing, and based on this comment and the following comment we have revised this sentence (see our response to the following comment 6). 

In addition, we revised the last section before the Strenghts and limitations section:

“However, when finalizing this study we were unaware of studies based on random samples among the general population published in peer-reviewed journals, on the perceived risks, the use of preventive measures and their pre-outbreak determinants, to compare our findings with.”

6. This statement, " peer-reviewed population-based studies assessing the perceived risks of corona infection, measures and their determinants are absent." is incorrect. The authors need to highlight the following landmark longitudinal study that also assessed perceived risk and determinants. Please mention the findings of this study in the Introduction.

Wang C, Pan R, Wan X, et al. (2020) A Longitudinal Study on the Mental Health of General Population during the COVID-19 Epidemic in China [published online ahead of print, 2020 Apr 13]. Brain Behav Immun. 2020; S0889-1591(20)30511-0. doi:10.1016/j.bbi.2020.04.028

Response

We understand the comment of the reviewer, but we could not refer to this interesting study because it was published online after we submitted our manuscript. 

Based on this comment and comment 5 we revised this section therefore as follows: 

“The study of Wang and colleagues [8], using a snowball sampling strategy in mainland China with surveys at the end of January and the end of February 2020, showed that 11.2% (first survey) and 9.1% (second survey) did find it very likely contracting COVID-19 during the pandemic. In addition, 11.9% (first survey) and 8.9% (second survey) did find it not very likely or not likely at all surviving if infected by COVID-19. Both variables were associated with current anxiety or depression symptoms. In total, 59.8% (first survey) and 73.2% (second survey) did always wear facemasks regardless of the presence or absence of symptoms; 66.6% (first survey) and 73.9% (second survey) did always wash hands after touching contaminated objects. The frequency of used preventive measures was negatively associated with current mental health problems. However, to the best of our knowledge, to date prospective studies conducted among random samples of the general population assessing the perceived risks of corona infection, preventive measures taken and pre-outbreak determinants of perceived risks and measures taken, are absent. Aim of the present prospective study, based on a random sample of the general population, is to shed light on this gap of scientific knowledge”.

In the discussion section we added:

In the study by Wang and colleagues [8] about 10% did not found it very likely or not likely at all to survive COVID-19. We have no data to compare these findings with.

7. Under discussion, the authors need to have a global view and need to discuss findings beyond Germany and Canada.

Response

We are not sure if we correctly understand this comment. We refer to studies conducted in Germany, Canada, United States and China (the last country after the revision) to compare our findings with (we are unaware of similar studies conducted in, for example, France and the UK). Our Future research and Conclusions sections, as well as remarks in the Strengths and limitations section, are not limited to any country.

8. For example, the authors found that "Multivariable logistic regression analyses showed that respondents with pre-outbreak anxiety and depression symptoms more often perceived the risk becoming ill when infected as high. Please refer to the following studies and discuss the challenges faced by psychiatric patients with anxiety and depression during COVID-19 lockdown.

Hao F, Tan W, Jiang L, et al. Do psychiatric patients experience more psychiatric symptoms during COVID-19 pandemic and lockdown? A Case-Control Study with Service and Research Implications for Immunopsychiatry [published online ahead of print, 2020 Apr 27]. Brain Behav Immun. 2020;S0889-1591(20)30626-7. doi:10.1016/j.bbi.2020.04.06

Response

We appreciate this suggestion very much, but this paper of the reviewer is beyond the aim of the present study.

9. Under discussion, the authors stated "Another limitation is that we not were able to include children. It is unknown to what extent children’s perceptions of the risks and the measures they taken resembles those of adults and especially parents and other family members." Please refer to the following study from China that included participants as young as 12 year old. Students were more affected due to disruption of academic studies:

Wang C, Pan R, Wan X, et al. (2020) A Longitudinal Study on the Mental Health of General Population during the COVID-19 Epidemic in China [published online ahead of print, 2020 Apr 13]. Brain Behav Immun. 2020; S0889-1591(20)30511-0. doi:10.1016/j.bbi.2020.04.028

Response

We do not understand the added value of referring to the paper of the reviewer in this section because no information is provided about how children perceived the risks and preventive measures taken. 

10. The authors should add one additional limitation. There was no mention of occupation of participants. the authors should discuss the impact of COVID on general workforce and healthcare professionals. 

Please discuss the findings of the following studies:

Tan W, Hao F, McIntyre RS, et al. Is Returning to Work during the COVID-19 Pandemic Stressful? A Study on Immediate Mental Health Status and Psychoneuroimmunity Prevention Measures of Chinese Workforce [published online ahead of print, 2020 Apr 23]. Brain Behav Immun. 2020;S0889-1591(20)30603-6. doi:10.1016/j.bbi.2020.04.055

Chew NWS, Lee GKH, Tan BYQ, et al. A multinational, multicentre study on the psychological outcomes and associated physical symptoms amongst healthcare workers during COVID-19 outbreak [published online ahead of print, 2020 Apr 21]. Brain Behav Immun. 2020;S0889-1591(20)30523-7. doi:10.1016/j.bbi.2020.04.049

 Response

We fully agree with the reviewer that employment status is an important variable to consider in predicting perceived risks and preventive measures taken. Based on this comment and comment 4 of reviewer 1, we therefore re-analyzed our data by adding employment status to the list of predictors in the multivariable logistic regression analyses. 

For the revised tables we refer to the revised manuscript (because of the lengthy tables) and for the revised text in the manuscript we refer to our response to comment 4 of reviewer 1. 

Our study focused on the general population, and not on specific groups in the workforce such as healthcare workers. We therefore do not consider this as a limitation, although we agree that it is a very interesting question. We believe that, as suggested by the reviewer, by adding employment status to the list of predictors we improved our manuscript within the scope of our focus on the general population. 

We nevertheless briefly referred in the discussion section to the interesting study of the reviewer as follows.

“Finally, it was beyond the scope of the present study to assess perceived risks and preventive measures taken, as well as its pre-outbreak predictors, among (specific groups of) the workforce when returning to work after a lockdown. For this purpose, we refer to the study of Tan and colleagues [22].”

---

## [Decision Letter · Decision Letter 1]

1 Jun 2020

Pre-outbreak determinants of perceived risks of corona infection and preventive measures taken. A prospective population-based study

PONE-D-20-09023R1

Dear Dr. van der Velden,

We are pleased to inform you that your manuscript has been judged scientifically suitable for publication and will be formally accepted for publication once it complies with all outstanding technical requirements.

With kind regards,

Geilson Lima Santana, M.D., Ph.D.

Academic Editor

PLOS ONE

Additional Editor Comments (optional):

Reviewers' comments:

Reviewer's Responses to Questions

**Comments to the Author**

1. If the authors have adequately addressed your comments raised in a previous round of review and you feel that this manuscript is now acceptable for publication, you may indicate that here to bypass the “Comments to the Author” section, enter your conflict of interest statement in the “Confidential to Editor” section, and submit your "Accept" recommendation.

Reviewer #1: All comments have been addressed

Reviewer #2: All comments have been addressed

2. Is the manuscript technically sound, and do the data support the conclusions?

Reviewer #1: Yes

Reviewer #2: Yes

3. Has the statistical analysis been performed appropriately and rigorously? 

Reviewer #1: Yes

Reviewer #2: Yes

4. Have the authors made all data underlying the findings in their manuscript fully available?

Reviewer #1: Yes

Reviewer #2: Yes

5. Is the manuscript presented in an intelligible fashion and written in standard English?

Reviewer #1: Yes

Reviewer #2: Yes

6. Review Comments to the Author

Reviewer #1: Authors had answered my all proposals and questions well, Please accept and publish online as soon as possible.

Reviewer #2: I recommend acceptance. Thank you for amendments and I am happy with the amendments. The journal will go ahead with publication.

7. PLOS authors have the option to publish the peer review history of their article (what does this mean?). If published, this will include your full peer review and any attached files.

Reviewer #1: No

Reviewer #2: Yes: Roger Ho

---

## [Editor Report · Acceptance letter]

17 Jun 2020

PONE-D-20-09023R1 

Pre-outbreak determinants of perceived risks of corona infection and preventive measures taken. A prospective population-based study 

Dear Dr. van der Velden:

I'm pleased to inform you that your manuscript has been deemed suitable for publication in PLOS ONE. Congratulations! Your manuscript is now with our production department. 

Kind regards, 

on behalf of

Dr. Geilson Lima Santana 

Academic Editor

PLOS ONE